# Soil depth determines the microbial communities in *Sorghum bicolor* fields within a uniform regional environment

Emily R. Murray,[1,2] Jeremiah J. Minich,[1] Jocelyn Saxton,[3] Marie de Gracia,[3] Nathaniel Eck,[3] Nicholas Allsing,[1] Justine Kitony,[1] Kavi Patel-Jhawar,[1] Eric E. Allen,[2] Todd P. Michael,[1,2] Nadia Shakoor[3]

**ABSTRACT** *Sorghum bicolor,* an important global crop, adapted to thrive in hotter and drier conditions than maize or rice, has deep roots that interact with a stratified soil microbiome that plays a crucial role in plant health, growth, and carbon storage. Microbiome studies on agricultural soils, particularly fields growing *S. bicolor*, have been mostly limited to surface soils (<30 cm). Here we investigated the abiotic factors of soil properties, field location, depth, and the biotic factors of sorghum type across 38 genotypes of the soil microbiome. Utilizing 16S rRNA gene amplicon sequencing, our analysis reveals significant changes in microbial composition and decreasing diversity at increasing soil depths within *S. bicolor* fields, regardless of genotype or field, with microbial richness and diversity declining to a minimum at the 60–90 cm layer and increasing beyond the 90 cm depth. Notably, specific microbial families, such as Thermogemmatisporaceae and an unclassified family within the ABS-6 order, were enriched in deeper soil layers beyond 30 cm. These findings highlight the importance of soil depth in agricultural soil microbiome studies.

**IMPORTANCE** *Sorghum bicolor* is a valuable model for studying the microbiome in deep soils, which is crucial for enhancing carbon sequestration in agricultural systems. As we look to crops with deeper roots for improved carbon storage, it is essential to move beyond the traditional focus on surface soils in agricultural settings. This study shifts that focus by investigating microbial dynamics at greater soil depths, revealing significant changes in microbial composition and diversity with increasing depth, revealing the critical role of deep-soil microbiomes in nutrient cycling and carbon sequestration in agricultural fields with the deep-rooted crop *S. bicolor*. By exploring these processes beyond surface soils, this research supports the development of sustainable agricultural practices that can better harness the potential of deep-rooted crops for long-term carbon storage.

**KEYWORDS** soil microbiome, soil depth, *Sorghum bicolor*, crop, 16S rRNA sequencing, microbial diversity, agricultural microbiology, depth-related microbial communities, root mass, soil organic carbon

S*orghum bicolor* (L) Moench ranks as the fifth most produced grain worldwide and is considered a primary food source for 750 million people in the arid and semi-arid tropics of Africa, Asia, and Latin America (1). There are various sorghum utilization types, including grain, forage, cellulosic, sweet, and energy, which are used for food (grain and sweet), animal feed (forage), fuel (energy and cellulosic), and building materials (cellulosic). *S bicolor* is commonly produced by small-scale subsistence farmers growing traditional varieties with limited access to fertilizers, pesticides, improved seeds, good soil, and water. *S. bicolor* is often cultivated in conditions that can be too hot or dry for *Zea mays* (Maize) or *Oryza sativa* (Rice) (2–4). Its resilience to poor-growing conditions

**Peer Reviewer** Azdayanti Muslim, Universiti Teknology Mara, Sungai Buloh, Selangor, Malaysia

Address correspondence to Nadia Shakoor, NShakoor@danforthcenter.org, or Todd P. Michael, tmichael@salk.edu.

T.P.M. is a co-founder of Cquesta, a company that works on crop root growth and carbon sequestration. The other authors have nothing to declare.

See the funding table on p. 17.

makes it an important crop in these regions, which are also susceptible to changing precipitation patterns due to climate change (5).

One adaptation of *S. bicolor* to dry and hot conditions is deep roots, enabling access to water that can remain in deep layers of the soil during dry periods (6, 7). In addition, deep roots have been shown to uptake certain nutrients such as calcium and potassium better than shallow roots (8) and can access surplus nitrogen in deep soil (7), with some deep-rooted species being capable of utilizing nitrate-N ($NO_3$-N) stores from deep layers of soil (9). Roots of *S. bicolor* occurring below 30 and 60 cm have been shown to affect above-ground yield significantly (6) and can reach maximum depths of 150 cm (10), 190 cm (11, 12), 200 cm deep in clay-rich soil, and >240 cm deep in sandy soils (13).

In addition, *S. bicolor* deep roots are important for soil organic carbon levels and longer-term carbon sequestration (14–16). Deep roots have been shown to have a slower decomposition rate compared to roots closer to the surface (17). In a 7-year trial focusing on bioenergy *S. bicolor*, Shahandeh and colleagues (18) demonstrated increased soil organic carbon within the deepest section of the measured soil profile (60–90 cm) (18). This rise occurred even though root biomass accumulation at this depth is significantly lower than in the top 20 cm of the soil profile.

Root depth can be impacted by several factors, one of which is the properties of the soil and the plant's ability to penetrate various soil types (19). Deep soil tends to have high compaction and clay content, leading to higher density and lower porosity compared to surface soil, making it more difficult for plant roots to penetrate (20). The nutrient content is stratified in the soil as well; for example, phosphorus and potassium tend to have higher concentrations in surface soils, while sodium, chloride, and magnesium tend to have higher concentrations in deep layers, allowing deep roots to have access to different nutrients (21–23).

The stratification of chemical and physical properties with depth has also been shown to be crucial factors influencing the community composition and functional properties of the soil microbiome (23–28). Microbes cycle nutrients in the soil and associate with plants in the area surrounding plant roots, known as the rhizosphere, while having a significant influence on plant health, growth, development, and evolutionary adaptations (29). Physical properties, such as the high density and low porosity of deep soil, can occlude microbes, limiting motility and preventing microbes from interacting with plants, while chemical properties alter the availability of nutrients for microbes. Even within aggregates, the chemical composition can be different from surrounding soils, supporting a distinct microbial community (23). Importantly, rhizosphere microbiome trends are not the same among all plants and depend on the individual plant properties and root architecture (19, 30).

Different species and genotypes of plants produce unique profiles of root exudates and volatile organic compounds that attract a plant growth-promoting microbial community (31–33). Several studies have investigated the rhizosphere microbiome of field-grown *S. bicolor*, with variations influenced by aforementioned factors such as soil properties, genotype, and plant developmental stage in the top 15, 20, or 30 cm of soil or without reporting depth (34–39). A core microbiome *S. bicolor* was reported by Abera et al. (34) from samples collected from 48 fields across a 2,000 km transect through the Ethiopian sorghum belt, with 267 amplicon sequence variants (ASVs) belonging to three families, accounting for 0.8% of the total ASVs. These were Rubrobacteriaceae 52.8%, Pseudomonadaceae 27.0%, and Beijerinckiaceae 20.2% (0–20 cm depth) (34). Furthermore, studies have investigated specific *S. bicolor* genotype-microbiome interactions within the top layers of soil (34, 40–43). For example, Abera et al. (34) found distinct recruitment of Pseudomonadaceae by the stay-green, drought-tolerant, and wild *S. bicolor* genotypes (0–20 cm depth)(34). Campolino et al. (40) found that genotype significantly impacted the beta diversity of the microbiome rhizosphere (40). In another study, Lopes et al. (41) grouped *S. bicolor* genotypes by "N-stress sensitive" (N108B, N110B, Theis, and Northern Sugarcane) or "N-stress tolerant" (Macia, N109B, and Rancher) (41). They found a significant difference in the sensitive vs. tolerant

genotype groups' rhizosphere microbiome communities in a high N-field, with 25 taxa altered in relative abundance. In addition, Schempler et al. (2018) found that the *S. bicolor* genotype SRN-39 promoted a stronger co-variance between bacterial and fungal communities compared to other genotypes (43). Another study by Schempler et al. (2017) showed that in the late stages (days 35 and 50), the *S. bicolor* genotype had a small but significant effect on the bacterial community composition (42).

The deep roots of *S. bicolor* and its associated microbiome play a critical role in nutrient uptake, above-ground biomass yield, and carbon sequestration capabilities. Despite research on the *S. bicolor* rhizosphere microbiome, including investigations into genotype effects, nutrient availability, soil properties, and developmental stages, existing studies have not explored soil microbiomes associated with *S. bicolor* beyond 30 cm. This research aims to fill this knowledge gap by investigating the influence of soil depth on microbial community composition and diversity in *S. bicolor* fields. Specifically, this study aims to characterize microbial communities across six depths (0–150 cm) using 16S rRNA sequencing, identify depth-dependent changes in microbial diversity and composition across different *S. bicolor* utilization types and genotypes, as well as explore the relationships between soil properties, root mass, and microbial community structure. In addition, we extend the comparison of our data to other studies on the *S. bicolor* microbiome. We hypothesize that soil depth is the dominant factor shaping microbial community composition and diversity, with deeper soils exhibiting reduced microbial diversity and distinct community structures compared to surface soils.

## MATERIALS AND METHODS

### Sorghum planting experimental design

Two experimental fields, designated A3 and D5 were established in O'Fallon, Missouri, USA (38.8477, −90.68664 and 38.8494,− 90.68742)(Fig. 1A). O'Fallon, Missouri, located in the Midwestern United States, experiences a humid subtropical climate (Köppen climate classification: Cfa) characterized by hot and humid summers and cool to mild winters. The annual mean temperature is approximately 13.6°C, with average summer highs around 26.2°C in July and winter lows averaging −0.3°C in January. The region receives an average annual precipitation of about 1,059 mm. Relative humidity varies throughout the seasons but generally averages around 64%–68%. In addition, Missouri has approximately 170 frost-free days per year, with the average last frost occurring in mid-April and the first frost typically arriving in late October, defining the region's growing season. O'Fallon experiences approximately 200 sunny days per year, with peak solar radiation in the summer months. Conversely, it records approximately 107 rainy days annually (https://en.climate-data.org/north-america/united-states-of-america/missouri/o-fallon-16780/, https://www.britannica.com/place/Missouri-state/Climate).

The fields A3 and D5 described in this study were approximately 100 meters apart. Both fields share the same soil type, Hurst silt loam. There were no data on soil variations between these fields prior to the study, and no other significant variations were observed.

The fields were planted with *S. bicolor* between June 3rd and June 17th, 2021. Field A3 was planted with 21 genotypes covering five utilization types: grain, sweet, cellulosic, forage, and energy. Field D5 was planted with 17 genotypes covering three utilization types: grain, sweet, and forage (Fig. 1B). The fields contained two to three replicates per genotype randomized within the field, with no overlap of genotypes between fields. Nitrogen fertilizer was applied at a rate of 400 lbs/acre before planting. Soil core collection was conducted at the end of the growing season in October 2021, after the sorghum plants had matured, approximately 4 months after planting (June 3rd–June 17th, 2021). This timing allowed for a comprehensive assessment of the plants' impact on soil properties and microbial communities after the entire growth cycle. In Missouri, October typically experiences more stable temperatures than the summer months, reducing the potential influence of daily temperature fluctuations on soil microbial communities.

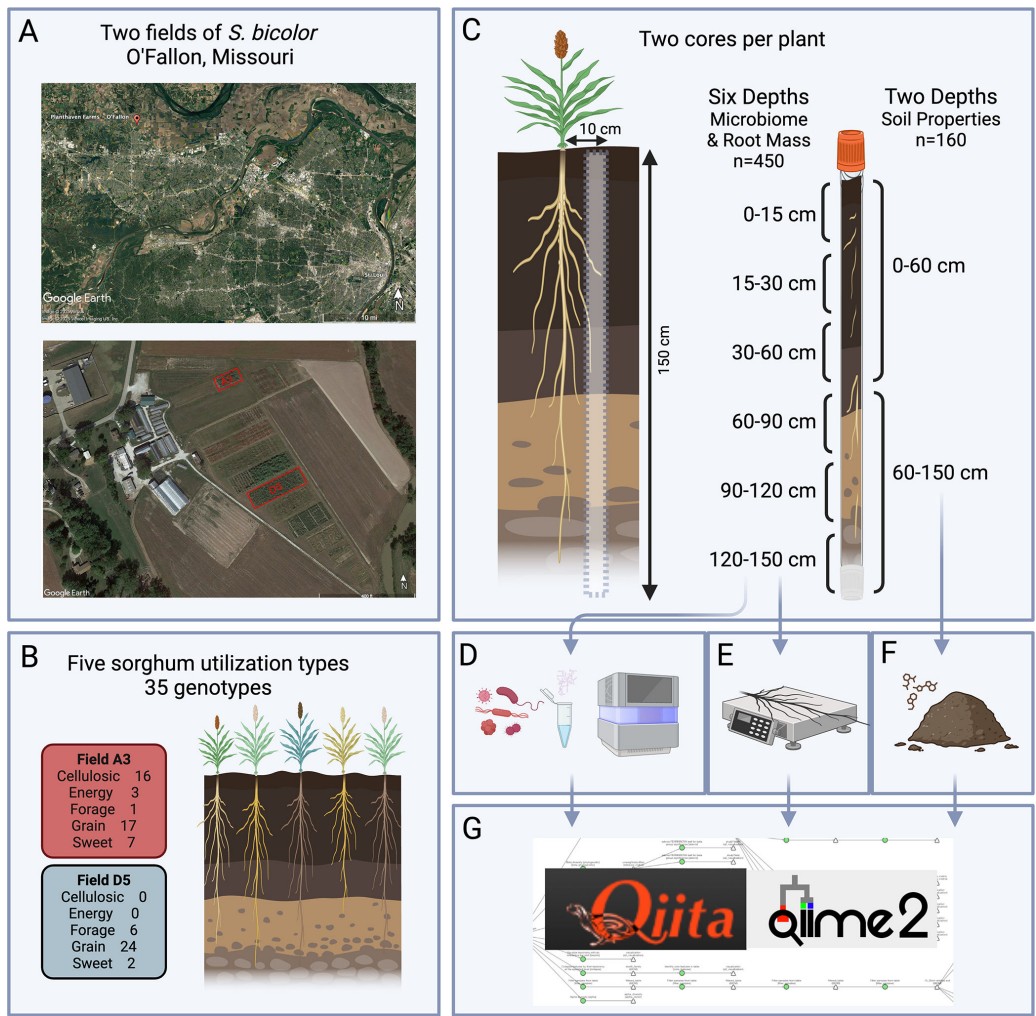

**FIG 1** Experimental approach to soil microbiome analysis in *S. bicolor* fields. (A) Two fields were planted in O'Fallon, Missouri. (B) Covering five *S. bicolor* utilization types and 35 genotypes. (C) Two soil cores were taken per plant, ~10 cm from the base of the plant, 150 cm deep. Cores were split into six depths for microbiome and root analysis and two depths for soil property analysis. (D) 16S V4 rRNA sequencing was done on *n* = 450 soil samples. (E) Roots were weighed. (F) Soil properties (*n* = 160) were analyzed for pH, soluble salts, organic matter, nitrate-nitrogen, phosphorus, potassium, calcium, magnesium, sodium, sulfur, zinc, iron, manganese, copper, total N and P, inorganic N and P, organic N, P, and C. (G) Data were combined and analyzed in Qiita and Qiime 2. This figure was created using BioRender (https://biorender.com/).

Sampling was completed within 1 week, ensuring all samples were collected under consistent environmental conditions to enhance comparability across plots and genotypes. Bulk soil was collected from 150-cm deep soil cores extracted 7–10 cm away from the plant stalk and above-ground roots using a GeoProbe coring machine. Two cores 7.6 cm in diameter were collected per plot, with genotypes replicated 1–3 times (Fig. 1C).

At the end of each sampling day, soil cores were transported to the laboratory by automobile, with transport times typically under 1 hour. Upon arrival, the samples were immediately placed in a cold room maintained at 4°C. The core washing process was completed within 1 month, during which time the cores were consistently maintained at 4°C. Studies have shown that short-term storage at this temperature has minimal impact on microbial community composition, preserving diversity and ensuring suitability for downstream analysis (44–46). Each core was carefully segmented into six predetermined depth sections (0–15, 15–30, 30–60, 60–90, 90–120, and 120–150 cm). From each depth section, aliquots of approximately 100 mg were prepared and

immediately stored at −80°C after processing to preserve sample integrity while allowing for efficient processing.

## Soil property measurements

Soil samples were composited into two layers, shallow (0–60 cm) and deep (60–150 cm), resulting in 160 samples, which were subsequently sent to Ward Laboratories, Inc. (Kearny, NE) for soil property analysis. Soil health assessments included measurements of pH, soluble salts, and organic matter. In addition, concentrations of nitrate-nitrogen, phosphorus, potassium, calcium, magnesium, sodium, sulfur, zinc, iron, manganese, and copper were determined. The Haney Test was used to evaluate total nitrogen and phosphorus, both in their inorganic and organic forms, soil respiration (1-day CO2-C), water extractable organic carbon and nitrogen, P2O5 and K2O, wet aggregate stability, and total organic carbon (47)(Fig. 1F). A table with *S. bicolor* genotype, utilization type, root, and soil characteristics is found in Table S1.

## Soil microbiome processing and root biomass phenotyping

Soil samples from the cores were segmented for analyses at six predetermined depths: 0–15, 15–30, 30–60, 60–90, 90–120, and 120–150 cm. This segmentation facilitated microbiome profiling and root biomass assessment across 450 samples. Approximately 100 mg of the soil-root mixture was sent to the company Microbiome Insights, which performed extraction and sequencing of the 16S rRNA V3-V4 regions using the 515F-806R primers following Earth Microbiome Project protocols (48, 49) (Fig. 1D). Root biomass was quantitatively determined by washing, drying, and weighing the roots (Fig. 1E). Data were combined and analyzed in Qiita and Qiime2 (50, 51) (Fig. 1G).

## Data processing

All data were uploaded to the microbiome multi-omics study platform Qiita (51) as per-sample FASTQ files, excluding barcodes (Qiita ID: 14937). The demultiplexing process was executed using the 'Split Libraries Fastq' command, setting the Phred quality score offset to 33. Following this, all sequences were trimmed to achieve a consistent length of 150 base pairs. Utilizing the Deblur "reference hit" (52) and the Greengenes_13.8 database (53), the taxonomic classification of the microbial communities was conducted using the BIOM feature table (54) generated from sequencing data. Greengenes was selected over alternative databases due to its integration capabilities, having unified genomic and 16S rRNA data into a single, comprehensive reference tree, facilitating seamless integration and comparison across different data types, including amplicon and shotgun metagenomic data sets, providing enhanced accuracy and consistency of taxonomic assignments (55). The use of Greengenes ensures that the data set is compatible with future analyses that may integrate other data types, thereby enhancing the utility and reproducibility of the findings. The classification was performed employing QIIME2's (50) pre-fitted sklearn-based taxonomy classifier, facilitating taxonomy assignment to each feature present in the table. Samples were rarefied to a depth of 5,000 sequences per sample. Rarefaction retained 387 of 450 samples (86.00%) and 1,935,000 of 6,789,276 (28.50%) features (Fig. S1).

Alpha and beta diversity analyses were performed to assess the diversity of the *S. bicolor*-associated soil microbiome (56). Alpha diversity was assessed for all samples at varying depths using Shannon Entropy (richness and abundance) (57), Faith's Phylogenetic Distance (richness, weighted by the phylogeny) (58), and observed features (richness). Diversity metrics were generated in Qiita, and the relationship between soil depth and alpha diversity was evaluated using Spearman's rank correlation (59). This non-parametric correlation coefficient was selected to measure the strength and direction of association between soil depth and alpha diversity metrics. In addition, to explore the impact of sorghum utilization type on alpha diversity at specific soil depths, the Kruskal-Wallis (60) test was applied separately for each diversity metric at each depth level. We then performed Spearman's rank correlation analysis to determine

the associations between diversity indices and 54 environmental variables (Table S2). All statistical analyses were conducted using Python's SciPy library, with a significance level set at $P < 0.001$.

Beta diversity was assessed using weighted and unweighted UniFrac distance metrics (61). A PERMANOVA test was employed to evaluate the significance of observed group differences in beta diversity (62). The BIOM table was filtered to 15 cm (0–15 cm), 30 cm (15–30 cm), 60 cm (30–60 cm), 90 cm (60–90 cm), 120 cm (90–120 cm), or 150 cm (120–150 cm) to isolate the effects of sorghum utilization type, genotype, and total phosphorus at each specific depth. Each depth-specific BIOM table was then analyzed with Adonis PERMANOVA using unweighted and weighted UniFrac distances.

The differential abundance of microbial taxa at the ASV level across various soil depths was assessed using the Analysis of Composition of Microbiomes with Bias Correction (ANCOM-BC2) in R (63), which corrects for common biases in microbiome data analysis. The ASV-level classification table was merged with the corresponding metadata and transformed into a phyloseq object utilizing the phyloseq package in R (64). The ANCOM-BC2 analysis was executed with the ancombc2 function from the ANCOMBC package, specifying the depth as the grouping variable. Results were filtered to retain significant values ($P$-value $< 0.05$) and plotted as a heatmap (Fig. S3). The results were further filtered to contain taxa with a log fold change (LFC) higher than 3 in a pairwise comparison or be the most enriched or depleted in any pairwise comparison, resulting in 18 taxa. These 18 taxa were then plotted as a heatmap across all depths for an in-depth analysis.

All measured soil properties were analyzed for distribution of values within the two depth groups (shallow: 0–60 cm and deep: 60–150 cm), and the Kruskal-Wallis H-test was employed. This non-parametric test is suitable for comparing more than two independent groups and does not assume a normal distribution of the data.

## Meta-analysis

We performed a search of PubMed for articles published before 5 July 2023 using the following search string: sorghum AND (root OR rhizosphere OR soil) AND (microbiome OR metabolome) AND (16S OR Metagenome). We examined the titles and abstracts of all returned citations and reviewed selected full-text articles based on our inclusion/exclusion criteria.

## Inclusion and exclusion criteria

Studies were selected if they met these criteria following the Earth Microbiome Project protocol (49): (i) utilized high-throughput sequencing methods to quantify microorganisms in the root or rhizosphere of *S. bicolor* grown in soil, (ii) utilized PCR amplification of the 16S rRNA V3-V4 region using primers 515F-806R or 515F-909R (48), and (iii) extracted DNA using MoBio power soil DNA isolation kit OR MP fastDNA spin kit. We excluded review papers, meta-analyses, abstracts or conference proceedings, and articles with duplicate data. After the revision and filtering process and checking for data availability, two studies were retained (65, 66) (ncbi.nlm.nih.gov/sra/?term = SRP110648, https://www.ncbi.nlm.nih.gov/bioproject/?term=PRJNA723704).

## Data processing

All raw data were downloaded as FASTQ files from the National Center for Biotech Information (NCBI) Sequence Read Archive (SRA) database. A study was created in the microbiome multi-omics study platform Qiita, and metadata and preparation information were prepared for each study. Metadata documents were created using metadata files from the NCBI SRA database and supplementary information provided in the paper. A preparation information file was generated using the sample name and the run prefix from the raw data files. The preparation files, metadata files, and raw data were uploaded to the respective Qiita studies for subsequent analysis (Qiita IDs: 15230, 15207).

Data were uploaded and processed in the same way as described in the single study analysis, with the exception that all samples were rarefied to a depth of 1,332 sequences per sample to prevent entire study dropout, as some of the studies had very low sequencing depth. Rarefaction retained 747.252 distinct features, representing 10.00% of the total observed features, across 561 samples, which accounted for 93.19% of the total sample cohort. Following rarefaction, filtering removed samples categorized as "bulk soil" and those lacking pH metadata.

To assess core and dominant features at varying depths, the feature table was filtered to produce separate tables for depths 0–30 cm, classified as surface soil, and 30 cm, classified as subsurface soil. Microbial features were then aggregated at the family taxonomic level (level 5). "Identify core features" was processed at each depth and analyzed at 100% to identify core features for all samples, surface soil, and subsurface soil independently. Dominant features were identified by the most abundant taxa present in at least 50% of samples in each group.

## RESULTS

### Soil properties and root mass are significantly different between shallow (0–60 cm) and deep (60–150 cm) soil

Soil samples from two experimental fields in O'Fallon, Missouri, were divided into shallow (0–60 cm) and deep (60–150 cm) layers for comprehensive analysis, including pH, soluble salts, organic matter, and nutrient concentrations (Fig. 1). We examined the variation in soil properties and root mass across different soil depths in two fields (A3 and D5). The analyzed properties included measurements of nitrogen, phosphorus, carbon, metals, and other measures of soil properties (Fig. 2A and B; Table S3). In both fields, root mass, soil health, and carbon properties, including CO2-C and Total WEOC, were lower in the deep soil, while nitrogen and metals exhibited variable trends between the two layers. Phosphorus properties trends between depths differed across fields, being reduced in the deep soil in field A3 and increased in the deep soil in field D5.

Nitrogen levels exhibited variable trends across soil depths in both fields. In field A3, total nitrogen content decreased by 36.47%, from an average of 7.5 ppm in the shallow soil to 4.8 ppm in the deep soil (Kruskal-Wallis H: 132.28, $P < 0.0001$). Field D5 showed a similar pattern, with total nitrogen levels lower by 20.82%, from 7.61 ppm in the shallow soil to 6.03 ppm in the deep soil (Kruskal-Wallis H: 37.84, $P < 0.0001$). Other nitrogen properties, such as total organic nitrogen and nitrate, were also reduced in the deep soil across both fields. However, ammonium was increased in the deep soil in both fields (Fig. 2C, Table S3).

For carbon properties, the total water-extractable organic carbon (WEOC) showed a marked decrease with depth in both fields. In field A3, WEOC decreased by 35.42% from an average of 66.02 ppm in the shallow soil to 42.64 ppm in the deep soil (Kruskal-Wallis H: 45.00, $P < 0.0001$). Similarly, in field D5, WEOC levels dropped by 41.00%, from 70.88 ppm in the shallow soil to 41.84 ppm in the deep soil (Kruskal-Wallis H: 94.99, $P < 0.0001$). In addition, carbon dioxide-carbon (CO2-C) levels were consistently lower in the deep soil across both fields (Fig. 2D; Table S3)

Phosphorus properties exhibited contrasting trends between fields A3 and D5. In field A3, plant available phosphorus levels significantly decreased by 51.15%, from an average of 100.63 ppm in the shallow soil to 49.15 ppm in the deep soil (Kruskal-Wallis H: 47.28, $P < 0.0001$). Conversely, in field D5, plant available phosphorus levels increased by 36.29%, from 94.34 ppm in the shallow soil to 128.59 ppm in the deep soil (Kruskal-Wallis H: 47.28, $P < 0.0001$) (Fig. 2E; Table S3)

Several metals showed significant variations between shallow and deep soil layers in both fields. In field A3, potassium levels increased by 50.10%, from an average of 40.25 ppm in the shallow soil to 60.41 ppm in the deep soil (Kruskal-Wallis H: 43.21, $P < 0.0001$). Similarly, in field D5, potassium levels rose by 55.32%, from 45.67 ppm in the shallow soil to 70.94 ppm in the deep soil (Kruskal-Wallis H: 117.41, $P < 0.0001$). Magnesium levels also exhibited an increase of 33.48%, from 102.56 ppm in the shallow soil to 137.00 ppm

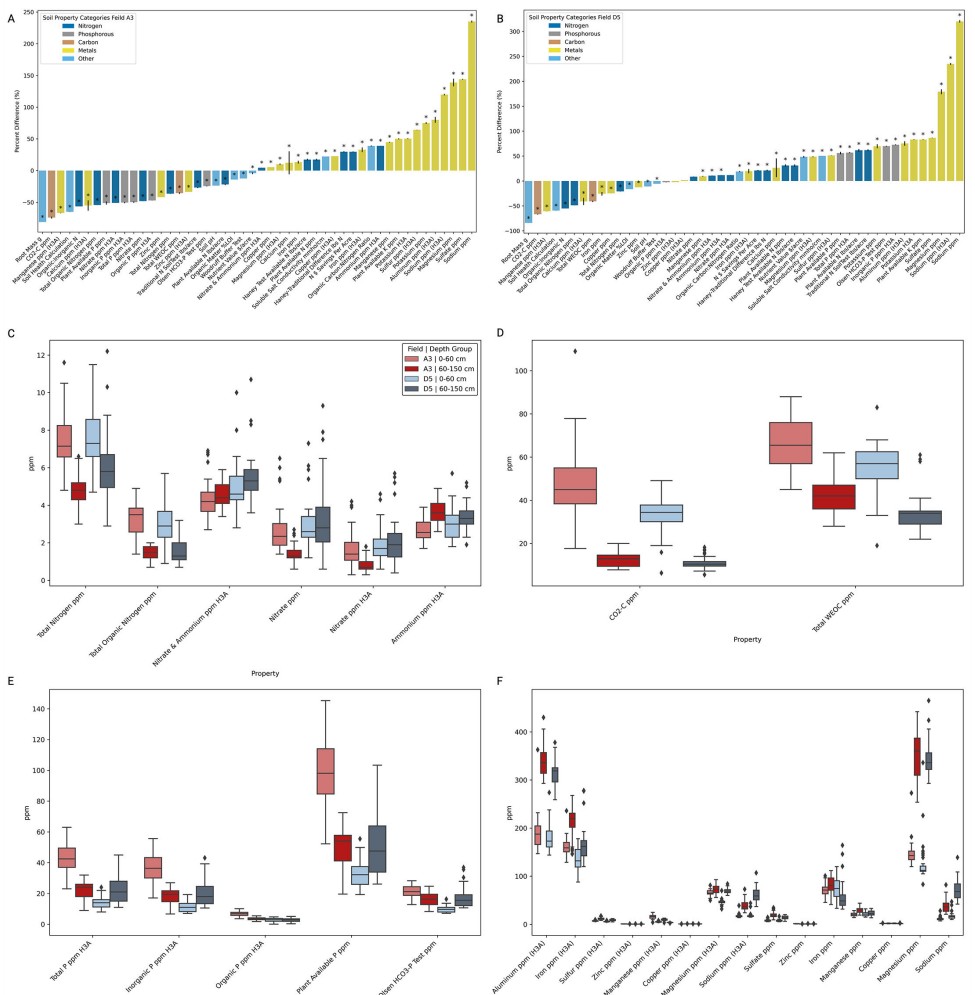

**FIG 2** Differences in soil properties and root mass between shallow (0–60 cm) and deep (60–150 cm) soil depths. Percent differences in various soil properties and root mass (g) between shallow (0–60 cm) and deep (60–150 cm) soil depths in fields (A) A3 and (B) D5. The properties analyzed include measurements of nitrogen (dark blue), phosphorus (gray), carbon (orange), metals (yellow), and other relevant soil health indicators (light blue). The bars represent the percent change calculated as [(mean value at 60–150 cm − mean value at 0–60 cm)/mean value at 0–60 cm] × 100. The soil properties are sorted by the magnitude of their percent differences, facilitating easy comparison. Positive values indicate higher levels in deeper soils, while negative values indicate higher levels in shallower soils. The error bars represent the combined standard error of the mean values for each depth group. (C-F) Boxplots of soil properties measured in ppm are grouped into four categories: (C) nitrogen, (D) carbon, (E) phosphorous, and (F) metals, comparing shallow (0–60 cm) and deep (60–150 cm) soil layers by field (A3, D5). Nitrogen properties include total nitrogen, total organic nitrogen, nitrate & ammonium (H3A), nitrate, and ammonium (H3A). Carbon properties cover CO2-C and total WEOC. Phosphorous properties include total P (H3A), inorganic P (H3A), organic P (H3A), plant available P, and Olsen HCO3-P test. Metal properties cover aluminum, iron, sulfur, zinc, manganese, copper, magnesium, sodium, and sulfate, measured using the H3A method and other methods—statistics in Table S3.

in the deep soil in field A3 (Kruskal-Wallis H: 29.48, *P* < 0.0001), and a 40.21% increase from 110.45 ppm in the shallow soil to 154.81 ppm in the deep soil in field D5 (Kruskal-Wallis H: 67.56, *P* < 0.0001). In addition, not shown in Fig. 2 box and whisker plots, due to extremely high concentrations, calcium levels also increased with depth, showing a 42.97% rise from 1456.67 ppm in the shallow soil to 2081.51 ppm in the deep soil in field A3 (Kruskal-Wallis H: 38.75, *P* < 0.0001), and a 35.14% rise from 1592.48 ppm in the shallow soil to 2152.84 ppm in the deep soil in field D5 (Kruskal-Wallis H: 82.34, *P* < 0.0001). However, a few metals, such as zinc, were an exception, showing a significant

increase in the shallow soil. In field A3, zinc levels increased by 28.57%, from 1.40 ppm in the deep soil to 1.80 ppm in the shallow soil (Kruskal-Wallis H: 25.34, $P < 0.0001$); similarly, in field D5, zinc levels increased by 24.18%, from 1.58 ppm in the deep soil to 1.96 ppm in the shallow soil (Kruskal-Wallis H: 21.47, $P < 0.0001$) (Fig. 2F; Table S3).

Soil health was reduced in deep soil. In field A3, soil health decreased by 37.75%, from 7.63 in the shallow soil to 4.75 in the deep soil (H: 40.54, $P < 0.0001$). Field D5 showed a similar trend, with soil health declining by 32.89%, from 8.12 in the shallow soil to 5.45 in the deep soil (H: 85.23, $P < 0.0001$). In addition, root mass varied with depth, sorghum utilization type, and genotype (Fig. 3). On average, root mass decreased significantly in both fields, with field A3 showing an 81.10% reduction from 0.53 g in the shallow soil to 0.10 g in the deep soil (H: 50.68, $P < 0.0001$) and field D5 showing a 75.81% reduction from 0.62 g in the shallow soil to 0.15 g in the deep soil (H: 100.01, $P < 0.0001$).

## Soil depth determines the microbial communities in *S. bicolor* fields

Spearman's rank correlation analyses revealed a significant negative correlation between soil depth and each alpha diversity metric (Faith's phylogenetic diversity, Shannon's entropy, and the observed features), indicating a strong trend of decreasing microbial diversity with depth. The correlation coefficients were −0.740 ($P < 0.001$) for Faith's phylogenetic diversity (PD), −0.784 ($P < 0.001$) for Shannon's entropy, and −0.765 ($P < 0.001$) for observed features. However, despite the overall trend of decreasing diversity with depth, a minimum in alpha diversity was observed at the 60–90 cm layer. Decreases in diversity with depth were consistent across all sorghum utilization types (Fig. 4A through C).

Alpha diversity increased with both root mass and total organic carbon across all samples; however, when controlling for soil depth, the correlations ceased to be significant, suggesting that the observed relationships between alpha diversity, root mass, and organic carbon are depth-dependent (Fig. S2). All soil properties were analyzed, and depth had a stronger influence over alpha diversity than any soil property (Table S2). Given that depth was the strongest variable driving the microbiome, all further analyses were conducted while controlling for depth.

Depth also exhibited the strongest significant impact on community composition; both unweighted and weighted UniFrac analyses showed strong significant effects (Pseudo-F statistic (unweighted) = 40.063, $P < 0.001$; Pseudo-F statistic (weighted) = 155.691, $P < 0.001$). Field also showed a significant and expected impact on community composition (Pseudo-F statistic (unweighted) = 15.744, $P < 0.001$; Pseudo-F statistic (weighted) = 20.640, $P < 0.001$). Both soil depth and field are key drivers of microbial diversity in these environments (Fig. 4D through H, Table 1), with microbial taxa showing differences in relative abundance by depth (Fig. 4J).

By contrast, unweighted and weighted analyses yielded non-significant results for sorghum utilization type and genotype, indicating that neither sorghum utilization type nor the genetic variation within the *S. bicolor* significantly influenced the microbial community structure across all samples (Fig. 4F and I, Table 1). Significant interactions between field and depth were observed (Pseudo-F statistic [unweighted] = 7.394, $P < 0.001$; Pseudo-F statistic [weighted] = 10.417, $P < 0.001$), highlighting the complex interplay between spatial factors in shaping microbial communities. However, interactions involving sorghum utilization type and genotype with depth did not significantly affect the microbial communities, suggesting that the effect of depth on microbial diversity is not modulated by these factors (Table 1).

Further analysis revealed distinct patterns in microbial community composition across different soil depths within each field. The impact of field depth on community composition remained when examining specific soil layers, with the strongest effect from fields observed in the upper layers. For instance, at soil depths of 0–15 cm and 15–30 cm, both unweighted and weighted UniFrac analyses indicated robust field effects (Pseudo-F statistic [unweighted] = 12.192 and 12.215, $P < 0.001$; Pseudo-F statistic [weighted] = 23.363 and 28.636, $P < 0.001$, respectively)(Table 1).

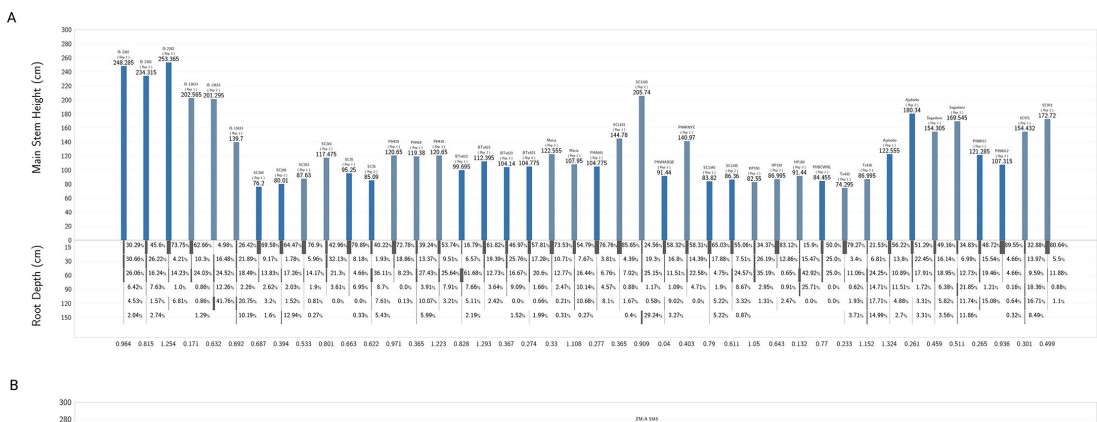

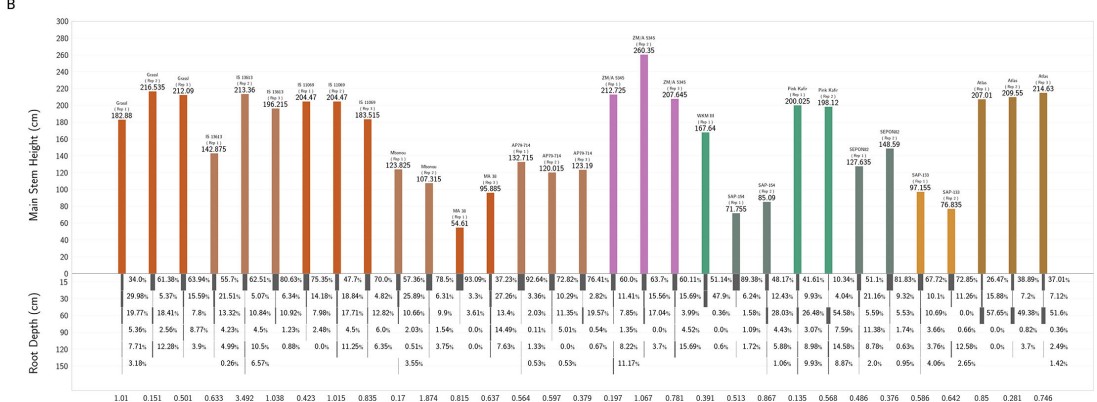

**FIG 3** Root depth distribution and above-ground height (cm) for individual plants and genotypes across five sorghum utilization types: (A) grain (blue), (B) cellulosic (red), energy (purple), forage (green), and sweet (brown). Root mass percentage is shown for six depth intervals: 0–15 cm, 15–30 cm, 30–60 cm, 60–90 cm, 90–120 cm, and 120–150 cm. Total root mass (g) is annotated below each plant's root profile.

As soil depth increased to 30–60 cm, 60–90 cm, and beyond, the influence of field conditions on microbial community structure remained significant but showed a decreasing trend. At soil depths of 30–60 cm and 60–90 cm, the Pseudo-F values were lower (Pseudo-F statistic [unweighted] = 6.789 and 6.116, $P < 0.001$; Pseudo-F statistic [weighted] = 8.734 and 13.971, $P < 0.001$, respectively), suggesting a diminishing influence of surface-derived factors with increased depth. Notably, the effect continued to wane in the deeper layers (90–120 cm and 120–150 cm), although the impact remained significant except for weighted unifrac in the deepest level (Pseudo-F statistic [unweighted] = 5.036 and 2.931, $P < 0.001$; Pseudo-F statistic [weighted] = 13.316 and 2.451, $P < 0.001$ and $P = 0.025$, respectively)(Table 1).

Since phosphorus properties differ between fields A3 and D5, with a significant decrease in deep soil in A3 but a significant increase in deep soil in D5, we examined how these trends influenced microbial community structure. Compared to depth and field, total phosphorus had a much weaker effect (Pseudo-F statistic [unweighted] = 7.05, $P < 0.001$; Pseudo-F statistic [weighted] = 12.33, $P < 0.001$). To control for depth and assess how these changes influence microbial populations in the deep soil, each layer was analyzed independently. Significant interactions were identified only in the 15–30 cm layer based on unweighted unifrac analysis (Pseudo-F statistic [unweighted] = 2.34, $P < 0.001$), suggesting that other environmental factors may overshadow the direct effects of phosphorus in this study (Table S5).

Our analysis revealed that depth was the primary driver of microbial community composition in this study, followed by field, with significant interactions between these two factors, with field effects being the strongest in the surface layers of soil. By contrast, sorghum utilization type and genotype did not significantly influence the overall microbial communities.

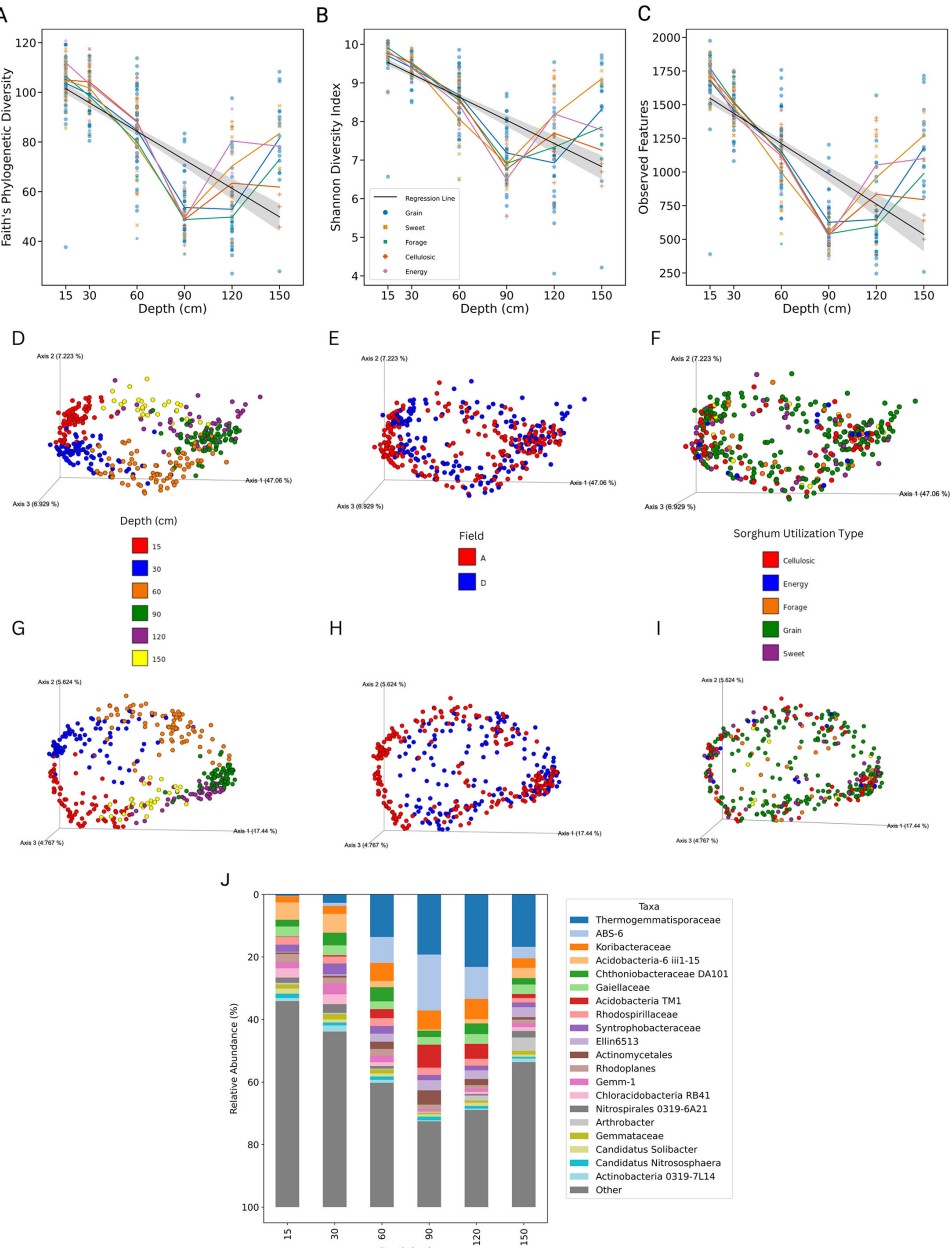

**FIG 4** Influence of soil depth on microbial communities in *S. bicolor* fields**.** This figure illustrates the variations in alpha diversity across different soil depths and the corresponding microbiome community composition. Alpha diversity metrics, including (A) Faith's PD, (B) Shannon index, and (C) observed features, are plotted against soil depth, regression lines shown in black, and each sorghum utilization type is represented as line plots. The x-axis represents soil depth, while the y-axis corresponds to the respective alpha diversity metric. Principal coordinate analysis (PCoA) demonstrates the microbiome community composition variation using unweighted and weighted UniFrac metrics. The top row displays the (D–F) unweighted UniFrac results, and the bottom row presents the (G–I) weighted UniFrac results. Each column represents a different factor: (D, G) sorghum type, (E, H) field, and (F, I) depth, with factors color-coded by type. Clustering within these PCoA plots visualizes the similarity among microbiome samples. (J) Relative abundance of the top 20 bacterial species labeled with the lowest common ancestor across different soil depths, with each taxon represented by a unique color. The data are presented as stacked bars, where each bar's height corresponds to the proportion of a taxon's abundance relative to the total bacterial community at that depth. For labeling simplicity, the 0–15 cm layer is shortened to 15 cm; the 15–30 cm layer is shortened to 30 cm; the 30–60 cm layer is shortened to 60 cm, the 60–90 cm layer is shortened to 90 cm, the 90–120 cm layer is shortened to 120 cm, and the 120–150 cm layer is shortened to 150.

**TABLE 1** Summary of PERMANOVA based on the ADONIS method evaluating the effects of field, soil depth, sorghum type, and genotype[a]

| Factor | Unweighted UniFrac | | Weighted UniFrac | |
|---|---|---|---|---|
| | Pseudo-F | *P* | Pseudo-F | *P* |
| Field | 15.7440541 | 0.001*** | 20.63962602 | 0.001*** |
| Depth | 40.0627793 | 0.001*** | 155.6905594 | 0.001*** |
| Sorghum type | - | 0.828 | - | 0.875 |
| Genotype | - | 0.988 | - | 0.988 |
| Field: depth | 7.39403044 | 0.001*** | 10.41686957 | 0.001*** |
| Depth: sorghum type | - | 0.908 | - | 0.558 |
| Depth: genotype | - | 0.909 | - | 0.67 |
| Field 0–15 cm | 12.191863 | 0.001*** | 23.362931 | 0.001*** |
| Field 15–30 cm | 12.214975 | 0.001*** | 28.636058 | 0.001*** |
| Field 30–60 cm | 6.789346 | 0.001*** | 8.733691 | 0.001*** |
| Field 60–90 cm | 6.116424 | 0.001*** | 13.970964 | 0.001*** |
| Field 90–120 cm | 5.035859 | 0.001*** | 13.316436 | 0.001*** |
| Field 120–150 cm | 2.930753 | 0.001*** | - | 0.025 |

[a]***$P \leq 0.001$; "–" indicates $P > 0.001$ and therefore the corresponding Pseudo-F value is not shown due to lack of statistical significance.

## Specific microbial taxa have differential abundance at various depths

The abundance of specific microbial taxa varied significantly with soil depth. Differential abundance analysis conducted using ANCOM-BC2 identified hundreds of microbial ASVs exhibiting significant changes across different soil layers (Fig. S3). Several taxa stood out as having a large and significant impact on differential abundance. There were 18 ASVs that were particularly notable due to either having the largest LFCs (>3) or being the single most enriched or depleted in any pairwise comparison (Fig. 5A).

Eight of the 18 taxa were generally enriched in soil layers below 30 cm compared to those above 30 cm. The taxa include an unclassified ASV in the GAL-15 phylum, two unclassified ASVs in the Chloroflexi phylum (Thermogemmatisporales order, B12-WMSP1 order), two unclassified ASVs in the AD3 phylum (JG37-AG-4 class and ABS-6 class), one unclassified ASV in the Actinobacteria phylum (Micrococcales order), and two unclassified ASVs in the Acidobacteria phylum (TM1 class and DA052 class). Conversely, six of 18 taxa were generally depleted in soil layers below 30 cm compared to those above it, including three ASVs in the Proteobacteria phylum (*Geobacter sp.* and two unclassified ASVs both in the Betaproteobacteria class), one unclassified ASV in the Gemmatimonadetes phylum (Gemmatimonadetes class), one unclassified ASV in the Acidobacteria phylum (RB41 order), and one ASV in the Crenarchaeota phylum (*Candidatus Nitrososphaera gargensis*).

The 60–90 cm layer exhibited unique patterns of microbial abundance. An unclassified ASV in the Proteobacteria phylum (Erythrobacteraceae family) was strongly enriched in this layer compared to all other layers and was the most strongly enriched taxa when compared to the two neighboring layers, 30–60 cm and 90–120 cm. In addition, an unclassified ASV in the Nitrospirae phylum (0319–6A21 family) was significantly depleted in the 60–90 cm layer when compared to all other depths, with the strongest LFC of −3.43 compared to the 15–30 cm layer. Similarly, the 90–120 cm layer showed unique enrichment of an unclassified ASV in the Proteobacteria phylum (Sinobacteraceae family), which was more enriched in this layer than any other layer, and the most enriched taxa in this layer compared to other layers below 30 cm. Finally, comparing only the shallow soil layers, Candidatus Nitrososphaera SCA1145 was the most depleted taxon in the 15–30 cm layer, compared to the 0–15 cm layer.

The largest of the large LFCs (>4) belonged to three taxa: the unclassified ASVs in the AD3 phylum (ABS-6 order), the unclassified ASV in the Chloroflexi phylum (Thermogemmatisporales order), and the unclassified ASV in the Acidobacteria phylum (TM-1 class).

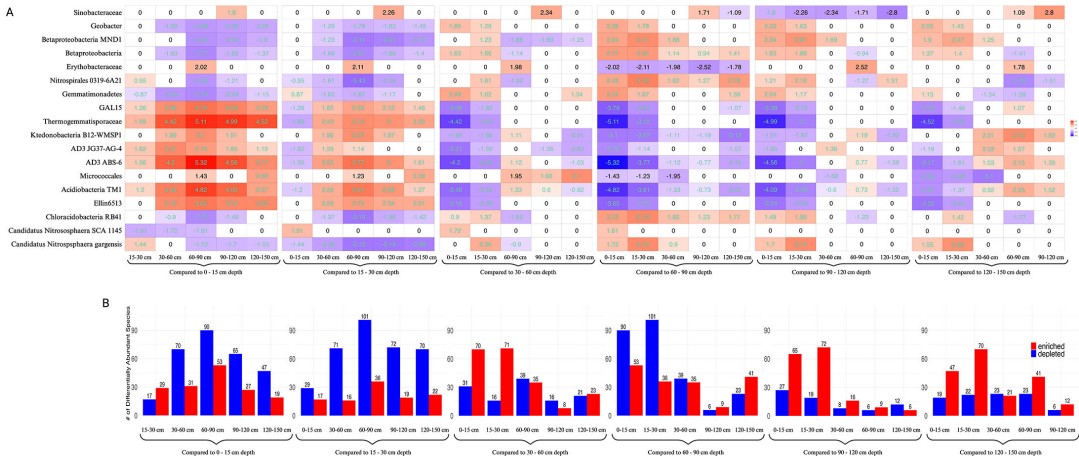

**FIG 5** Changes of microbial ASVs at various depths. (A) Heatmap showing the changes in the relative abundance of microbial ASVs labeled by lowest common ancestor in *S. bicolor* soil microbiomes at different depths, compared to a reference depth. The ASVs displayed have either a log fold change (LFC) greater than 3.0 or are the most enriched or depleted ASVs in any pairwise depth comparison. The y-axis lists taxa in alphabetical order, while the x-axis represents the depth comparisons. The color gradient ranges from blue (negative changes) to red (positive changes), with white indicating no significant change. The midpoint of the scale is 0, with a range of −5.4 to 5.4. Comparisons that passed the sensitivity analysis for pseudo-count addition are highlighted with aquamarine text. (B) Bar plots showing the total number of significantly enriched or depleted taxa for each reference depth. The y-axis represents the number of taxa, and the x-axis shows the depth comparisons. Enriched taxa (positive changes) are indicated in blue, while depleted taxa (negative changes) are shown in red, providing an overview of microbial shifts across depths.

All LFCs greater than four were strongly enriched in each depth group below 30 cm compared to the 0–15 cm depth (Fig. 5A).

Overall, general trends indicated distinct patterning between the top two soil layers compared to the bottom four layers. The 0–15 cm layer and the 60–90 cm layer exhibited the highest number of differentially abundant taxa, with 143 taxa either significantly enriched or depleted between them. By contrast, the 60–90 cm and 90–120 cm depths had the fewest differentially abundant taxa, with only 15 taxa showing significant changes (Fig. 5B).

## Meta-analysis enables core and dominant analysis while supporting soil depth as a primary driver of microbial community structure

The results of this study were integrated into a meta-analysis of existing studies on the *S. bicolor* microbiome to determine if a core soil microbiome existed for *S. bicolor*. The literature was searched, screened, and incorporated into the meta-analysis. Existing studies have only investigated the top 30 cm of soil, and when incorporated into a stacked taxa barplot and a PCoA plot, samples from the meta-analysis aligned with samples from the current study collected above 30 cm along the first principal coordinate (PCA1) in the PCoA plot (Fig. 6). PCA1 accounted for most of the variance in the microbial community data, supporting depth as a primary driver of microbial community structure. Depth was the strongest factor associated with microbiome composition (Adonis PERMANOVA Pseudo-F statistic [unweighted] = 38.564, $P < 0.001$; Pseudo-F statistic [weighted] = 143.481, $P < 0.001$) followed by experimental design [study] (Pseudo-F statistic [unweighted] = 2.536, $P < 0.001$; Pseudo-F statistic [weighted] = 99.277, $P < 0.001$). Effect sizes indicated that soil depth has a more pronounced influence than the experimental design (study) in shaping microbial assemblages, reinforcing that microbial diversity and distribution are strongly depth-dependent.

Core and dominant taxa analyses were conducted on the meta-analysis on all samples, surface soil (0–30 cm) and subsurface soil (60–150 cm). Rhodospirillaceae was found in 100% of all samples across three studies. An unclassified family in the Acidobacteria-6 class was present in 100% of surface soil samples (0–30 cm). The subsurface soil analysis was based on this study only because the additional studies in the meta-analysis

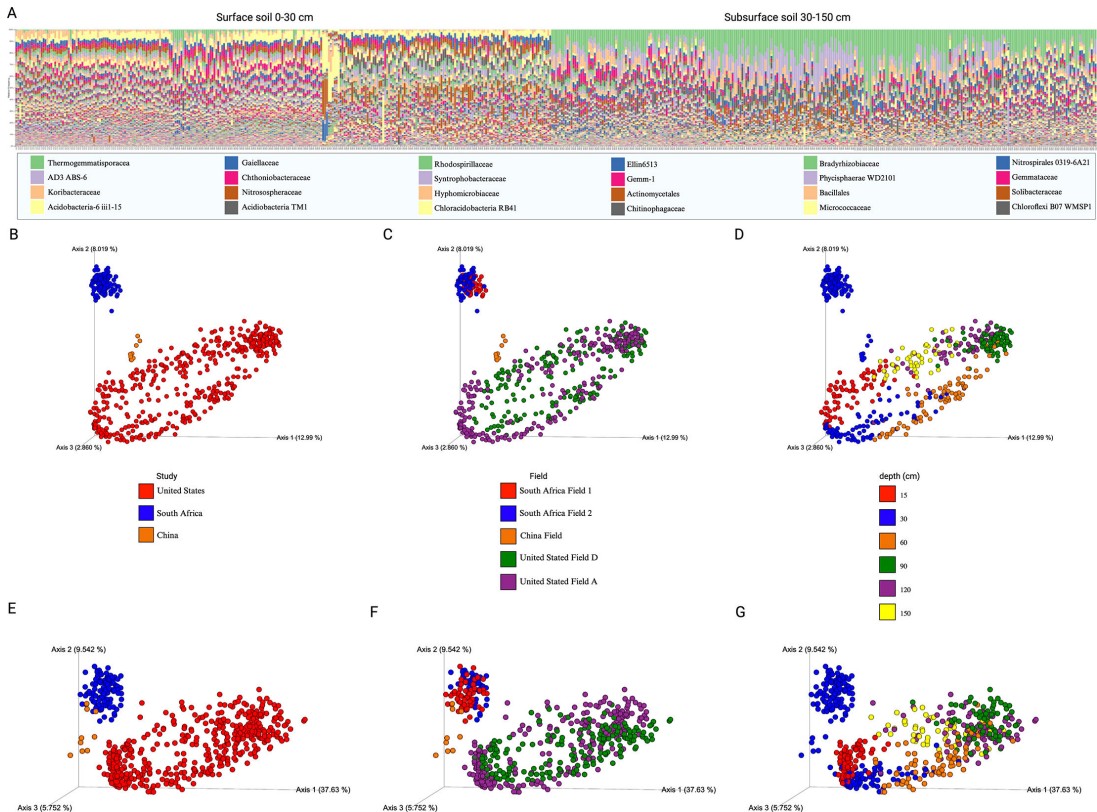

**FIG 6** Meta-analysis supports the influence of soil depth on microbial communities in *S. bicolor* fields. (A) Stacked bar plot of abundance of microbial families labeled with the lowest common ancestor for each sample, sorted by depth showing the community composition of surface vs. subsurface. Each taxon is represented by a unique color. The data are presented as stacked bars, where each bar's height corresponds to the proportion of a taxon's abundance relative to the total bacterial community for that sample. The legend shows the 20 most abundant families. (B–G) PCoA demonstrates the microbiome community composition variation using unweighted and weighted UniFrac metrics. The top row displays the (B–D) unweighted UniFrac results, and the bottom row presents the (E–G) weighted UniFrac results. Each column represents a different factor: (B, E) study, (C, F) field, and (D, G) depth, with factors color-coded by type. Clustering within these PCoA plots visualizes the similarity among microbiome samples. Samples from three data sets were collected from five fields distributed across three continents.

did not include samples deeper than 30 cm. In the subsurface soil, alongside Rhodospiril-laceae, the 10 most abundant taxa at the family level present in 100% of samples included Thermogemmatisporaceae, an unclassified family in the AD3 phylum (ABS-6 order), Koribacteraceae, two unclassified families in the Acidobacteria phylum (TM1 class and Ellin6513 order), Chthoniobacteraceae, Gaiellaceae, Syntrophobacteraceae, and Hyphomicrobiaceae (Table S4).

Dominant taxa were common (present in at least 50% of samples) and abundant. The top 5 most abundant families in over 50% of all samples were Thermogemmatispor-aceae, Koribacteraceae, unclassified Acidobacteria-6 class, Gaiellaceae, and Chthoniobac-teraceae. In the surface soil, the top 5 most abundant taxa in over 50% of samples were an unclassified family in the Acidobacteria-6 class, Nitrososphaeraceae, Gaiella-ceae, an unclassified family in the Acidobacteria phylum (RB41 order), and Hyphomi-crobiaceae. In the subsurface soil, the top 5 most abundant families in over 50% of samples were Thermogemmatisporaceae, unclassified family in the AD3 phylum (ABS-6 order), Koribacteraceae, Chthoniobacteraceae and unclassified family in the Acidobacte-ria phylum (TM1 class). Table S4 provides all core and dominant taxa.

## DISCUSSION

Depth is a critical factor in below-ground microbiome studies and is often neglected despite its profound impact on microbial communities. This study presents a comprehensive analysis of the microbial communities associated with *S. bicolor* at various soil depths, extending our understanding of soil microbiomes at depth in agricultural settings. Utilizing 16S rRNA amplicon sequencing, we documented significant changes in composition and diversity with increasing soil depth within *S. bicolor* fields, showing that including depth is integral to below-ground microbiome studies. This variability is visually evident in the community composition patterns displaying correlations between soil depth alpha diversity (Fig. 4A through C) and taxonomy bar plots (Fig. 3J and 5A). Notably, our primary data set reveals a significant similarity among samples from the same depth, as illustrated by the clustering observed in the PCoA plot (Fig. 4D through I). This pattern is supported by a meta-analysis of *S. bicolor* samples, which, despite their limitation to the top 30 cm of soil, align consistently with this study's shallow soil samples along the PCA1 axis of the PCoA plot (Fig. 6B through G). Oligotrophic, slow-growing taxa such as the Chloroflexi Thermogemmatisporaceae family, AD3 ABS-6 order, and Acidobacteria TM-1 class were more prevalent in deeper soil. Taxa belonging to the copiotrophic, fast-growing Proteobacteria phylum, as well as ammonia-oxidizing *C. gargensis* (67), were strongly enriched in the shallow soil. In addition, the genotype and utilization type of *S. bicolor* did not significantly influence the microbial community after controlling for depth and field, while the effects of root mass and soil organic carbon were determined to be depth-dependent.

Depth as a driving factor of the microbiome community aligns with previous research done in agricultural fields, including soy and corn (25), soy and wheat (68), rice (69), wheat (70), pomelo (71), and bioenergy crops (72), including the observation that alpha diversity generally decreases with depth (73). Despite the overall trend of diminishing diversity, we observed a minimum in alpha diversity in the 60–90 cm layer. This shift may indicate a disturbance of microbial communities at a stratification horizon and recovery beyond. The shallow soil contains higher levels of labile organic carbon and nutrients, favoring copiotrophic microorganisms that exhibit high growth rates, while the deep soils have reduced labile carbon, oxygen, and nutrients, but store recalcitrant carbon, favoring oligotrophic microorganisms with high affinity and slower growth rates that can outcompete faster-growing microorganisms in the deeper layer (74). Oligotrophic bacteria can decompose recalcitrant carbon pools dependent on nutrient availability, while deep-rooted crops can enhance carbon and transport essential nutrients from shallow soil to deep soil (75). A diverse ecosystem allows an ecosystem to adapt to a changing environment (76). Therefore, a thriving oligotrophic community in the deep soil may have important implications for the introduction of deep-rooted plants' effect on nutrient cycling and carbon sequestration.

Notably, half of the carbon sequestered in agricultural land is microbial necromass (77). Organic carbon originating from plants is metabolized by microbes, eventually becoming part of the microbial biomass. Following microbial cell death, lysis, and fragmentation, microbial cell compounds add to the microbial necromass pool (78). Microbial necromass becomes bound to clay mineral particles or stabilizes within micro-aggregates, also known as the entombment effect, entering the nonliving soil organic carbon pool and contributing to carbon sequestration (77, 79–81). The correct balance of carbon and nitrogen provided by decomposing plant roots increases the efficiency of microbial growth and provides a mechanism for carbon stabilization in the deep soil. Therefore, while plants may provide the initial carbon source, the role of microbial metabolism and accumulation beyond the rhizosphere has significant implications for carbon sequestration (82).

The influence of plants on the soil microbiome diminishes with distance from the root (83, 84). Previous studies have identified significant correlations between *S. bicolor* genotypes and the rhizosphere microbiome in the top layers of soil (34, 40–43). *S. bicolor* genotypes and utilization types are known to produce distinct exudates differentially,

which interact uniquely with microbial communities (32, 85). Our study did not find a correlation with *S. bicolor* genotype or utilization type because we analyzed bulk soil, which contained a mix of soil and root material, whereas prior studies analyzed rhizosphere samples, where plant influence is more pronounced. This distinction is important because plant type and exudate availability may be less critical when considering carbon sequestration in deep soil beyond the rhizosphere.

Some taxa, particularly in the top soil layers, were ubiquitous bacteria. Rhodospirillaceae was found in 100% of all samples, a type of non-sulfur purple photosynthetic bacteria, with 18 species and 17 of the 18 species capable of growth, with $N_2$ as the sole nitrogen source (86). This family of bacteria can grow photoautotrophically with light, carbon dioxide ($CO_2$), and either hydrogen gas ($H_2$) or hydrogen sulfide ($H_2S$). In addition, they can utilize organic acids, amino acids, sugars, alcohols, and aromatic compounds like benzoate and toluene as carbon sources (87). The most dominant member of the surface soils and present in every surface soil sample in the meta-analysis was an unclassified family in the Acidobacteria-6 class, which are globally distributed, highly metabolically diverse taxa that inhabit a wide variety of terrestrial and aquatic habitats (88). The following two most abundant families in the shallow soil were Gaiellaceae and an archaea, Nitrososphaeraceae, with *C. gargensis* being largely enriched in soil above 30 cm depth in the primary study; both are common taxa that are both obligately aerobic ammonia oxidizers. Living at the interface of soil and air and utilizing gases such as carbon dioxide ($CO_2$), hydrogen ($H_2$), nitrogen ($N_2$), and hydrogen sulfide ($H_2S$), as well as having the ability to metabolize complex biomolecules, highlights the critical functions of the soil's role in biogeochemical cycling, nitrogen fixation, and the breakdown of complex organic compounds (67, 89, 90).

The deep soil was characterized by oligotrophs, which are known to thrive in subsurface, carbon-limited environments (91). The most dominant family in the deep soil was Thermogemmatisporaceae. This family, first isolated from fallen leaves on geothermal soils, was characterized by Yabe et al. in 2011 as gram-positive and aerobic, forming branched mycelium with spores capable of oxidizing various carbon sources (92). One strain of Thermogemmatisporaceae is known to hydrolyze cellulose, with genome analysis revealing genes encoding 64 putative carbohydrate-active enzymes from 57 glycoside hydrolase families (93). The second most abundant taxa in the deep soil is an unclassified family in the ABS-6 order of the AD3 Phylum (renamed Dormibacteraeota). Dormibacteraeota are slow-growing aerobic heterotrophs that can survive in low-resource environments by storing and processing glycogen and trehalose. This candidate phylum also possesses type I and II carbon monoxide dehydrogenases, potentially enabling the use of trace amounts of carbon monoxide as an additional energy source (94). Previous studies have identified dark autotrophy, $CO_2$ fixation pathways, and iron and sulfur oxidation from bacterial genomes from deep soil (94), highlighting the unique metabolic capabilities of deep soil microbiota, their ability to thrive in low-resource environments, coupled with their slow growth rates and oligotrophic nature, giving them a vital role in carbon sequestration and nutrient cycling.

These findings have significant implications for agricultural practices, particularly in leveraging deep-soil microbiomes to enhance crop productivity and ecosystem sustainability. With substantial shifts in microbial composition with soil depth, specific taxa enriched in deep soil contribute differently to nutrient cycling and resource acquisition for deep-rooted crops. Integrating this knowledge into soil management strategies could lead to more precise fertilization practices, including the use of biofertilizers and reducing excessive inputs while optimizing nutrient availability for plant uptake (95). Furthermore, the capacity of deep-soil microbial communities to contribute to soil organic carbon stabilization presents opportunities for integrating carbon management into agricultural systems (96, 97). By understanding and leveraging microbial activity at deeper soil depths, agricultural practices can enhance nutrient availability and carbon sequestration potential and improve soil fertility.

In conclusion, soil depth is the primary factor determining microbial community structure. Therefore, as deep-rooted plants enhance carbon input into these layers, understanding the interactions and resilience of these microbial communities becomes crucial for developing sustainable agricultural practices and improving carbon sequestration strategies. Future research should include comparisons of the endosphere, rhizosphere, and bulk soil, incorporating bulk soil samples from field plots where *S. bicolor* is not grown, up to a depth of 150 cm, to elucidate differences between these environments. This research should also include field replication and temporal sampling. In addition, future studies should focus on deep sequencing and the development of metagenome-assembled genomes (MAGs) from under-characterized soil taxa at various depths to better understand the metabolic capabilities of these microbial communities, their interactions with specific plant exudates and carbon sources, and their adaptability to challenging conditions. This will help to understand their contributions to the carbon cycle and potential roles in carbon sequestration.

## ACKNOWLEDGMENTS

We thank the Michael lab for helpful conversations and input into this study. We thank Bryan Miles, Justin Allen, Jacob Stanton, Natalie Elam, Robert Lowery, Darren O'Brien, Bill Kezele, Cliff Moore, Avery Talgo, Dawn Reynolds, and the members of the Shakoor lab at the Donald Danforth Plant Science Center for their support in soil coring, sample processing, data collection, and curation. We also thank Wolfgang Busch for his supervision, team management, and contributions to this project.

This work was supported through the Salk Harnessing Plants Initiative (HPI) with funding from the TED Audacious, Bezos Earth Fund, Sempra Energy, and Hess Corporation. J.J.M. was supported by NSF Rules of Life (Award 2011004).

## AUTHOR AFFILIATIONS

[1]The Plant Molecular and Cellular Biology Laboratory, Salk Institute for Biological Studies, La Jolla, California, USA
[2]Scripps Institution of Oceanography, University of California San Diego, La Jolla, California, USA
[3]Donald Danforth Plant Science Center, St. Louis, Missouri, USA

## AUTHOR ORCIDs

Emily R. Murray  http://orcid.org/0009-0004-9349-4325
Jeremiah J. Minich  http://orcid.org/0000-0002-7202-965X
Jocelyn Saxton  http://orcid.org/0009-0007-4994-5389
Marie de Gracia  http://orcid.org/0000-0001-7426-8892
Nathaniel Eck  http://orcid.org/0000-0001-7417-1417
Nicholas Allsing  http://orcid.org/0000-0002-1145-1646
Justine Kitony  http://orcid.org/0000-0003-0355-9005
Kavi Patel-Jhawar  http://orcid.org/0009-0002-0644-3403
Eric E. Allen  http://orcid.org/0000-0002-1229-8794
Todd P. Michael  http://orcid.org/0000-0001-6272-2875
Nadia Shakoor  http://orcid.org/0000-0002-2035-7117

## FUNDING

| Funder | Grant(s) | Author(s) |
| --- | --- | --- |
| TED Audacious | | Todd P. Michael |
| Bezos Earth Fund | | Todd P. Michael |
| Sempra Energy | | Todd P. Michael |

| Funder | Grant(s) | Author(s) |
|---|---|---|
| Hess Corporation | | Todd P. Michael |
| NSF Rules of Life | 2011004 | Jeremiah J. Minich |
| Salk Harnessing Plants Initiative | | Nadia Shakoor |

## ADDITIONAL FILES

The following material is available online.

### Supplemental Material

**Supplemental figures (Spectrum02928-24-S0001.pdf).** Fig. S1 to S3.
**Table S1 (Spectrum02928-24-S0002.xlsx).** Sample metadata.
**Table S2 (Spectrum02928-24-S0003.xlsx).** Spearman's rank correlation coefficients between alpha diversity metrics and soil properties.
**Table S3 (Spectrum02928-24-S0004.xlsx).** Soil properties variation with depth, percent difference calculations, summary statistics, and Kruskal-Wallis statistics.
**Table S4 (Spectrum02928-24-S0005.xlsx).** Core and dominant microbial taxa across soil depths.
**Table S5 (Spectrum02928-24-S0006.xlsx).** Effects of phosphorus on microbial community structure using Unweighted and Weighted UniFrac distances.

### Open Peer Review

**PEER REVIEW HISTORY (review-history.pdf).** An accounting of the reviewer comments and feedback.

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
