## [Reviewer comments · Microbiology Spectrum]

Microbiology Spectrum

Soil depth determines the microbial communities in *Sorghum bicolor* fields within a uniform regional environment

Emily Murray, Jeremiah Minich, Jocelyn Saxton, Marie de Gracia, Nathaniel Eck, Nicholas Allsing, Justine Kitony, Kavi Patel-Jhawar, Eric Allen, Todd Michael, and Nadia Shakoor

Corresponding Author(s): Nadia Shakoor, Donald Danforth Plant Science Center

Review Timeline:

Submission Date:	November 13, 2024
Editorial Decision:	December 23, 2024
Revision Received:	March 3, 2025
Accepted:	March 7, 2025

Editor: Philips Akinwale

Reviewer(s): Disclosure of reviewer identity is with reference to reviewer comments included in decision letter(s). The following individuals involved in review of your submission have agreed to reveal their identity: Azdayanti Muslim (Reviewer #2)

Transaction Report:

DOI: <https://doi.org/10.1128/spectrum.02928-24>

Re: Spectrum02928-24 (Soil depth determines the microbial communities in Sorghum bicolor fields)

Dear Dr. Nadia Shakoor:

Thank you for the privilege of reviewing your work. Below you will find my comments, instructions from the Spectrum editorial office, and the reviewer comments.

Revision Guidelines

Sincerely,
Philip Akinwale
Editor
Microbiology Spectrum

Reviewer #1 (Comments for the Author):

General comments:

This paper investigates the role of soil depth in shaping microbial communities in agricultural soils. It focuses on Sorghum bicolor, which is a globally significant crop known for its resilience to heat and drought and its ability to develop deep root systems. The study emphasizes the importance of exploring deep soil microbiomes, which are often overlooked in agricultural research, and highlights their potential role in nutrient cycling and carbon sequestration. By analyzing microbial diversity and

community composition across six soil depths at 0-150 cm in two experimental fields, the authors provide a comprehensive examination of microbial dynamics in both shallow and deep soils. The study highlights the critical role of soil depth in shaping microbial communities and underscores the potential of deep-rooted crops in carbon sequestration. After reading through the manuscript, I found the findings are relevant to the aims and scope of the journal, and they made some contributions to agricultural microbiology. Therefore, I recommend accept for publication in *Microbiology Spectrum* after several aspects are clarified and improvements are made.

Specific comments:

Lines 149-150: rewrite aims and objectives as well as your hypothesis for this study.

Line 159: although there are two experimental fields (A3 and A5) selected, they are geographically very close. So, the study relies on a single geographical region of O'Fallon, Missouri, USA. This limits the generalizability of the findings to other agricultural systems or environmental conditions. Therefore, to avoid giving readers the false impression that this study is done at large geographical scales, the title should be changed to something like: Soil depth determines the microbial communities in two *Sorghum bicolor* fields.

Lines 166-169: the authors should add background info about the climate conditions such as annual mean temperature, humidity, climate pattern, number of sunny and raining days etc. These data are easily available on meteorological websites.

Lines 166-169: authors must clarify are the soil samples bulk soil or rhizosphere soil? Microbial composition is very different between these two soil environments.

Line 199: Greengenes database is outdated. Switching to a more recent database, such as SILVA, might improve taxonomic resolution and accuracy.

Line 209: please put rarefaction curves in the supplementary figure.

Lines 232-246: To further enhance the study, the authors should consider using SparCC (Sparse Correlations for Compositional data) to construct microbial co-occurrence networks, as it is particularly suited for handling the sparsity and compositional nature of microbiome data. Incorporating such a network-based approach would complement the current diversity and composition analyses. Please refer to the study by Tan et al. 2022 (Tan, H., Yu, Y., Zhu, Y., Liu, T., Miao, R., Hu, R., & Chen, J. (2022). Impacts of size reduction and alkaline-soaking pretreatments on microbial community and organic matter decomposition during wheat straw composting. *Bioresource Technology*, 360, 127549.) which demonstrated the utility of SparCC in revealing functional microbial interactions during organic matter decomposition.

Line 399: choose a more appropriate term for "Pseudo-F_unweighted"

Figure 2A and B: please rotate the soil property texts on the x axis by 45 degrees clockwise.

Figure 3A : enlarge the numbers.

Figure 4 A B and C: enlarge the numbers.

Figure 4J and Figure 5A and Figure 6A: This stacked bar plot is very messy with extra long OTU names. Annotate these OTUs using another newer database to species level and show these taxonomic names at species level.

Reviewer #2 (Comments for the Author):

This study explored on soil depth relationships with the microbial communities in *S. bicolor* fields and highlighted significant changes in microbial composition and decreasing diversity at increasing soil depths within *S. bicolor* regardless of genotype or fields. They also highlighted that specific microbial families, such as Thermogemmatosporaceae and an unclassified family within the ABS-6 order, were enriched in deeper soil layers beyond 30 cm. I found that this research has a proper study design and gives understanding how the dynamicity of the microbiome in deep-rooted crops of *S. bicolor* by highlighting the role of soil depth in shaping microbial communities and this could lead to development of sustainable agricultural practices that can better harness the potential of deep-rooted crops for long term carbon storage. There are so many variables that the author needs to carefully analyze individually and their association with the microbiota profiles. My specific comments are as follows:

Abstract

Lines 45-47 & 50-51 could be combined as both discuss alpha diversity, which increased beyond the 90 cm depth.

Methods

1. Could the authors specify the distance between fields A3 and D5? Any variation between these two fields?
2. Was the collection conducted during summer or another specific season? If so, could seasonal temperature variations significantly influence the microbial composition in the soil of *S. bicolor*?
3. Perhaps the author can describe briefly how the soil samples being transported to the laboratory. Were the samples kept at specific temperatures or preserved in a way to maintain microbial viability?

Results

4. The results mention differences in phosphorus properties between deep soils of fields A3 and D5 (increased significantly in deep soil of field A3, but increase significantly in deep soil of D5). Could the authors clarify why and how these changes in phosphorus trends influence the microbial population in the deep soil for each field?

Discussion

5. How could the result/knowledge from this study be applied to the improvement in agriculture practices? could the knowledge

in microbial diversity in deep soil layers guide soil management practices or fertilization strategies for *S. bicolor*?

Emily R. Murray
PhD Student
Email: emurray@salk.edu

**Salk Institute for Biological
Studies**
10010 N Torrey Pines Rd
La Jolla, California 92037
www.salk.edu

February 19, 2025

Philips Akinwole
Editor
Microbiology Spectrum

Dear Dr. Akinwole

Attached is our revised manuscript (Spectrum02928-24) of “**Soil depth determines the microbial communities in Sorghum bicolor fields,**” which we have submitted for publication in *Microbiology Spectrum*. We appreciate the reviewers’ constructive comments and the opportunity to revise the paper. We have carefully considered each point and have addressed all of them. Please see our explanation of the changes detailed in this letter below. The revised text in the manuscript is visible under markup.

Thank you for your continued consideration of this manuscript for publication in the *Microbiology Spectrum*.

Emily R. Murray
Ph.D. Student
The Salk Institute & Scripps Institution of Oceanography

Reviewer 1:

1. Lines 149-150: rewrite aims and objectives as well as your hypothesis for this study.

Response: We appreciate the reviewer's suggestion to clarify the aims, objectives, and hypothesis to improve the overall clarity and focus of the manuscript. Based on this feedback, we have revised the section in question to clearly articulate the study's aims, objectives, and hypothesis concisely and cohesively as follows:

“This research aims to fill this knowledge gap by investigating the influence of soil depth on microbial community composition and diversity in *S. bicolor* fields. Specifically, this study aims to characterize microbial communities across six depths (0-150 cm) using 16S rRNA sequencing, identify depth-dependent changes in microbial diversity and composition across different *S. bicolor* utilization types and genotypes, as well as, explore the relationships between soil properties, root mass, and microbial community structure. Additionally, we extend the comparison of our data to other studies on the *S. bicolor* microbiome. We hypothesize that soil depth is the dominant factor shaping microbial community composition and diversity with deeper soils exhibiting reduced microbial diversity and distinct community structures compared to surface soils.” (lines 201-209)

2. Line 159: although there are two experimental fields (A3 and A5) selected, they are geographically very close. So, the study relies on a single geographical region of O'Fallon, Missouri, USA. This limits the generalizability of the findings to other agricultural systems or environmental conditions. Therefore, to avoid giving readers the false impression that this study is done at large geographical scales, the title should be changed to something like: Soil depth determines the microbial communities in two Sorghum bicolor fields.

Response: We appreciate the reviewer's insight regarding the geographic scope of our study and the potential for misinterpretation of the findings' generalizability. To address this, we agree that the title should better reflect the study's specific context and limitations. Therefore, we have revised the title to emphasize the geographic specificity of the experimental fields while maintaining the focus on the central findings. The revised title is as follows:

Soil depth determines the microbial communities in *Sorghum bicolor* fields within a uniform regional environment (lines 2-3)

3. Lines 166-169: the authors should add background info about the climate conditions such as annual mean temperature, humidity, climate pattern, number of sunny and raining days etc. These data are easily available on meteorological websites.

Response: We appreciate the reviewer's suggestion to include background information on the climate conditions of O'Fallon, Missouri, to provide additional context for our study. We agree that this contextual information is crucial to understanding the environmental conditions under which the study was conducted. We have added a paragraph summarizing key climate characteristics of the region based on data from meteorological sources. Edited sentences read as follows:

“O’Fallon, Missouri, located in the Midwestern United States, experiences a humid subtropical climate (Köppen climate classification: Cfa) characterized by hot and humid summers, and cool to mild winters. The annual mean temperature is approximately 13.6°C, with average summer highs around 26.2°C in July and winter lows averaging -0.3°C in January. The region receives an average annual precipitation of about 1,059 mm. Relative humidity varies throughout the seasons but generally averages around 64-68%. Additionally, Missouri has approximately 170 frost-free days per year, with the average last frost occurring in mid-April and the first frost typically arriving in late October, defining the region’s growing season. O’Fallon experiences approximately 200 sunny days per year, with peak solar radiation in the summer months. Conversely, it records approximately 107 rainy days annually (<https://en.climate-data.org/north-america/united-states-of-america/missouri/o-fallon-16780/>, <https://www.britannica.com/place/Missouri-state/Climate>) (lines 214-234)

4. Lines 166-169: authors must clarify are the soil samples bulk soil or rhizosphere soil? Microbial composition is very different between these two soil environments.

Response: We appreciate the reviewer’s suggestion to clarify whether the soil samples were bulk soil or rhizosphere soil. This distinction has been added for clarity:

“Bulk soil was collected from 150 cm-deep soil cores extracted 7-10 cm away from the plant stalk and above-ground roots using a GeoProbe coring machine.” (lines 251-252)

5. Line 199: Greengenes database is outdated. Switching to a more recent database, such as SILVA, might improve taxonomic resolution and accuracy.

Response: We appreciate the reviewer’s suggestion to consider the SILVA database for improved taxonomic resolution and accuracy. However, we selected the updated version of the Greengenes database, as described in McDonald et al. (2024), because it offers a unique integration of genomic and 16S rRNA data into a comprehensive reference tree. The updated Greengenes database was chosen for its potential to facilitate seamless integration and comparison across diverse datasets, including amplicon and shotgun metagenomic data, in future studies.

This feature ensures that our findings remain compatible with broader analyses that may be conducted using this dataset in the future. Providing consistent and accurate taxonomic assignments aligns with the long-term goals of this research field. We have clarified this rationale in the manuscript to better explain the advantages of using Greengenes.

(Daniel McDonald, Yueyu Jiang, Metin Balaban, Kalen Cantrell, Qiyun Zhu, Antonio Gonzalez, James T. Morton, Giorgia Nicolaou, Donovan H. Parks, Søren M. Karst, Mads Albertsen, Philip Hugenholtz, Todd DeSantis, Se Jin Song, Andrew Bartko, Aki S. Havulinna, Pekka Jousilahti, Susan Cheng, Michael Inouye, Teemu Niiranen, Mohit Jain, Veikko Salomaa, Leo Lahti, Siavash Mirarab & Rob Knight. (2024). Greengenes2 unifies microbial data in a single reference tree. Nature Biotechnology).

The updated explanation in the manuscript is as follows:

“Greengenes was selected over alternative databases due to its integration capabilities, having unified genomic and 16S rRNA data into a single, comprehensive reference tree, facilitating seamless integration and comparison across different data types, including amplicon and shotgun metagenomic datasets, providing enhanced accuracy and consistency of taxonomic assignments (55). The use of Greengenes ensures that the dataset is compatible with future analyses that may integrate other data types, thereby enhancing the utility and reproducibility of the findings.” (lines 309-311)

7. Line 209: please put rarefaction curves in the supplementary figure.

Thank you for your suggestion to include the rarefaction curves in the supplementary materials. We have added these curves as a supplementary figure to provide additional context and to support the interpretation of sequencing depth and diversity metrics. The supplementary figure is labeled as Supplementary Figure 1, and we have referenced on line 315

8. Lines 232-246: To further enhance the study, the authors should consider using SparCC (Sparse Correlations for Compositional data) to construct microbial co-occurrence networks, as it is particularly suited for handling the sparsity and compositional nature of microbiome data. Incorporating such a network-based approach would complement the current diversity and composition analyses. Please refer to the study by Tan et al. 2022 (Tan, H., Yu, Y., Zhu, Y., Liu, T., Miao, R., Hu, R., & Chen, J. (2022). Impacts of size reduction and alkaline-soaking pretreatments on microbial community and organic matter decomposition during wheat straw composting. *Bioresource Technology*, 360, 127549.) which demonstrated the utility of SparCC in revealing functional microbial interactions during organic matter decomposition.

Response: We appreciate the reviewer’s suggestion to consider using SparCC to construct microbial co-occurrence networks. However, we chose not to incorporate SparCC or similar co-occurrence network analyses in this study due to concerns regarding the high rate of false positives associated with these methods, as highlighted in recent literature (Sophie Weiss, Will Van Treuren, Catherine Lozupone, Karoline Faust, Jonathan Friedman, Ye Deng, Li Charlie Xia, Zhenjiang Zech Xu, Luke Ursell, Eric J Alm, Amanda Birmingham, Jacob A Cram, Jed A Fuhrman, Jeroen Raes, Fengzhu Sun, Jizhong Zhou, Rob Knight, Correlation detection strategies in microbial data sets vary widely in sensitivity and precision, *The ISME Journal*) While such methods can be useful in certain contexts, their application requires careful consideration of the inherent limitations which can lead to inaccurate interpretations of ecological interactions. Given these limitations and the exploratory nature of our study, we have prioritized methods that provide robust and reliable insights into microbial diversity and community structure. We believe that our current analyses effectively address the primary goals of this research without introducing the potential pitfalls associated with co-occurrence networks.

Line 399: choose a more appropriate term for "Pseudo-F_unweighted"

Response: Thank you for pointing out the term "Pseudo-F_unweighted" To improve clarity and ensure the term is more intuitive for readers, we have replaced "Pseudo-F_unweighted" and "Pseudo-F_weighted." with "Pseudo-F statistic (unweighted)" and "Pseudo-F statistic (weighted)" in the manuscript.

Figure 2A and B: please rotate the soil property texts on the x axis by 45 degrees clockwise.

Response: Thank you for suggesting to rotate the soil property texts on the x-axis in Figure 2A and B. We agree this will improve readability. We have updated the figure accordingly, rotating the x-axis labels by 45 degrees clockwise to ensure the text is clear and easy to interpret.

Figure 3A : enlarge the numbers.

Response: Thank you for your suggestion to enlarge the numbers in Figure 3A. We agree enlarging the numbers will improve readability. We have updated the figure to ensure the numbers are larger and more readable.

Figure 4 A B and C: enlarge the numbers.

Response: Thank you for your suggestion to enlarge the numbers in Figure 4 A B C. We agree enlarging the numbers will improve readability. We have updated the figure to ensure the numbers are larger and more readable.

Figure 4J and Figure 5A and Figure 6A: This stacked bar plot is very messy with extra long OTU names. Annotate these OTUs using another newer database to species level and show these taxonomic names at species level.

Response: Thank you for your feedback regarding the stacked bar plots in Figures 4J, 5A, and 6A. We recognize the concern about the complexity of the OTU names and have revised the figures to display taxonomic labels at the lowest common ancestor (LCA) level as species-level annotation is not standard in 16S rRNA gene sequencing data.

Reviewer: 2

Abstract

Lines 45-47 & 50-51 could be combined as both discuss alpha diversity, which increased beyond the 90 cm depth.

Response: Thank you for your suggestion to combine lines 45-47 and 50-51 in the abstract, as they both discuss alpha diversity. We agree these can be combined and have revised the abstract

to integrate these lines into a single statement, improving clarity and flow. The revised text is as follows:

“Utilizing 16S rRNA gene amplicon sequencing, our analysis reveals significant changes in microbial composition and decreasing diversity at increasing soil depths within *S. bicolor* fields, regardless of genotype or field, with microbial richness and diversity declining to a minimum at the 60 - 90 cm layer and increasing beyond the 90 cm depth.” (lines 66-70)

Methods

1. Could the authors specify the distance between fields A3 and D5? Any variation between these two fields?

Response: Thank you for your question regarding the distance and variation between fields A3 and D5. We agree that this clarification adds value to the manuscript. The following text has been added to the manuscript for clarification:

“The fields A3 and D5 described in this study were approximately 100 meters apart. Both fields share the same soil type, Hurst silt loam. There are no data on soil variations between these fields prior to the study, and no other significant variations were observed.” (lines 235-237)

2. Was the collection conducted during summer or another specific season? If so, could seasonal temperature variations significantly influence the microbial composition in the soil of *S. bicolor*?

Response: Thank you for your question regarding the timing of soil collection and the potential influence of seasonal temperature variations on microbial composition. We agree that this is an important factor to address, as the timing and environmental conditions during sampling can significantly influence microbial community dynamics. To clarify, we have added the following text to the manuscript:

“Soil core collection was conducted at the end of the growing season in October 2021, after the sorghum plants had matured, approximately four months after planting (June 3rd-June 17th, 2021). This timing allowed for a comprehensive assessment of the plants’ impact on soil properties and microbial communities after the entire growth cycle. In Missouri, October typically experiences more stable temperatures than the summer months, reducing the potential influence of daily temperature fluctuations on soil microbial communities. Sampling was completed within one week, ensuring all samples were collected under consistent environmental conditions to enhance comparability across plots and genotypes.” (lines 243-251)

3. Perhaps the author can describe briefly how the soil samples being transported to the laboratory. Were the samples kept at specific temperatures or preserved in a way to maintain microbial viability?

Response: Thank you for your question regarding the transportation and preservation of soil samples. We agree that ensuring the proper handling and preservation of samples is critical for maintaining the integrity of microbial communities and generating reliable results. To address your point, we have added the following text to the manuscript to describe the transportation and preservation process in detail:

“At the end of each sampling day, soil cores were transported to the laboratory by automobile, with transport times typically under one hour. Upon arrival, the samples were immediately placed in a cold room maintained at 4°C. The core washing process was completed within one month, during which time the cores were consistently maintained at 4°C. Studies have shown that short-term storage at this temperature has minimal impact on microbial community composition, preserving diversity and ensuring suitability for downstream analysis (Edwards et al., 2024; Lauber et al., 2010; Rubin et al., 2013). Each core was carefully segmented into six predetermined depth sections (0-15, 15-30, 30-60, 60-90, 90-120, and 120-150 cm). From each depth section, aliquots of approximately 100 mg were prepared and immediately stored at -80°C after processing to preserve sample integrity while allowing for efficient processing. (lines 254-263)

Results

4. The results mention differences in phosphorus properties between deep soils of fields A3 and D5 (increased significantly in deep soil of field A3, but increase significantly in deep soil of D5). Could the authors clarify why and how these changes in phosphorus trends influence the microbial population in the deep soil for each field?

Response: Thank you for your question regarding the influence of phosphorus trends on microbial populations in the deep soils of fields A3 and D5. We agree that clarifying these relationships is important for understanding microbial community dynamics across depths. The following text was added/modified and supplemental table 5 was added:

“Since phosphorus properties differ between fields A3 and D5, with a significant decrease in deep soil in A3 but a significant increase in deep soil in D5, we examined how these trends influenced microbial community structure. Compared to depth and field, total phosphorus had a much weaker effect (Pseudo-F statistic (unweighted) = 7.05, $p < 0.001$; Pseudo-F statistic (weighted) = 12.33, $p < 0.001$). To control for depth and assess how these changes influence microbial populations in the deep soil, each layer was analyzed independently. Significant interactions were identified only in the 15-30 cm layer based on unweighted unifrac analysis (Pseudo-F statistic (unweighted) = 2.34, $p < 0.001$), suggesting that other environmental factors may overshadow the direct effects of phosphorous in this study (Supplementary Data 5). (lines 526-534)

5. How could the result/knowledge from this study be applied to the improvement in agriculture practices? could the knowledge in microbial diversity in deep soil layers guide soil management practices or fertilization strategies for *S. bicolor*?

Response: Thank you for your insightful question regarding the application of our findings to agricultural practices. We agree that the results of this study have important implications for improving soil management and fertilization strategies, particularly for deep-rooted crops like *Sorghum bicolor*. To address your point, we have added the following text to the Discussion section:

“These findings have significant implications for agricultural practices, particularly in leveraging deep-soil microbiomes to enhance crop productivity and ecosystem sustainability. With substantial shifts in microbial composition with soil depth, specific taxa enriched in deep soil contribute differently to nutrient cycling and resource acquisition for deep-rooted crops. Integrating this knowledge into soil management strategies could lead to more precise fertilization practices, including use of biofertilizers, reducing excessive inputs while optimizing nutrient availability for plant uptake (95). Furthermore, the capacity of deep-soil microbial communities to contribute to soil organic carbon stabilization presents opportunities for integrating carbon management into agricultural systems (96,97). By understanding and leveraging microbial activity at deeper soil depths, agricultural practices can enhance nutrient availability, carbon sequestration potential, and improve soil fertility.” (lines 748-758)

Re: Spectrum02928-24R1 (Soil depth determines the microbial communities in *Sorghum bicolor* fields within a uniform regional environment)

Dear Dr. Nadia Shakoor:

Your manuscript has been accepted, and I am forwarding it to the ASM production staff for publication. Your paper will first be checked to make sure all elements meet the technical requirements. ASM staff will contact you if anything needs to be revised before copyediting and production can begin. Otherwise, you will be notified when your proofs are ready to be viewed.

Sincerely,
Philips Akinwole
Editor
Microbiology Spectrum